# Ultrasound and Clinical Preoperative Characteristics for Discrimination Between Ovarian Metastatic Colorectal Cancer and Primary Ovarian Cancer: A Case-Control Study

**DOI:** 10.3390/diagnostics9040210

**Published:** 2019-12-01

**Authors:** Maciej Stukan, Juan Luis Alcazar, Jacek Gębicki, Elizabeth Epstein, Marcin Liro, Alexandra Sufliarska, Sebastian Szubert, Stefano Guerriero, Elena Ioana Braicu, Mariusz Szajewski, Małgorzata Pietrzak-Stukan, Daniela Fischerova

**Affiliations:** 1Department of Gynecologic Oncology, Gdynia Oncology Center, Pomeranian Hospitals, 81519 Gdynia, Poland; 2Department of Obstetrics and Gynecology, Clínica Universidad de Navarra, 31008 Pamplona, Spain; 3Department of Process Engineering and Chemical Technology, Faculty of Chemistry, Gdańsk University of Technology, 80233 Gdańsk, Poland; 4Department of Clinical Science and Education, Karolinska Institutet and Department of Obstetrics and Gynecology Södersjukhuset, 11883 Stockholm, Sweden; 5Department of Gynecology, Gynecologic Oncology and Gynecologic Endocrinology, Medical University, 80210 Gdańsk, Poland; 6Gynecologic Oncology Centre, Department of Obstetrics and Gynecology, First Faculty of Medicine, Charles University, General University Hospital in Prague, 12851 Prague, Czech Republic; 7Clinical Department of Gynecological Oncology, The Franciszek Lukaszczyk Oncological Center, 85796 Bydgoszcz, Poland; 82nd Department of Obstetrics and Gynecology, Centre of Postgraduate Medical Education, 01809 Warsaw, Poland; 9Department of Obstetrics and Gynecology, University of Cagliari, Policlinico Universitario Duilio Casula, Monserrato, 09124 Cagliari, Italy; 10Department of Gynecology, Campus Virchow, Charité - Universitätsmedizin Berlin, Freie Universität Berlin, Humboldt-Universität zu Berlin, and Berlin Institute of Health, 13353 Berlin, Germany; 11Department of Oncological Surgery, Gdynia Oncology Centre, 81519 Gdynia, Poland; 12Division of Propedeutics of Oncology, Medical University of Gdańsk, 80210 Gdańsk, Poland; 13Medicover, 80309 Gdansk, Poland

**Keywords:** ultrasound, ovary, neoplasm, colon, metastasic, model, risk prediction, cancer

## Abstract

The aim of this study was to describe the clinical and sonographic features of ovarian metastases originating from colorectal cancer (mCRC), and to discriminate mCRC from primary ovarian cancer (OC). We conducted a multi-institutional, retrospective study of consecutive patients with ovarian mCRC who had undergone ultrasound examination using the International Ovarian Tumor Analysis (IOTA) terminology, with the addition of evaluating signs of necrosis and abdominal staging. A control group included patients with primary OC. Clinical and ultrasound data, subjective assessment (SA), and an assessment of different neoplasias in the adnexa (ADNEX) model were evaluated. Fisher’s exact and Student’s *t*-tests, the area under the receiver–operating characteristic curve (AUC), and classification and regression trees (CART) were used to conduct statistical analyses. In total, 162 patients (81 with OC and 81 with ovarian mCRC) were included. None of the patients with OC had undergone chemotherapy for CRC in the past, compared with 40% of patients with ovarian mCRC (*p* < 0.001). The ovarian mCRC tumors were significantly larger, a necrosis sign was more frequently present, and tumors had an irregular wall or were fixed less frequently; ascites, omental cake, and carcinomatosis were less common in mCRC than in primary OC. In a subgroup of patients with ovarian mCRC who had not undergone treatment for CRC in anamnesis, tumors were larger, and had fewer papillations and more locules compared with primary OC. The highest AUC for the discrimination of ovarian mCRC from primary OC was for CART (0.768), followed by SA (0.735) and ADNEX calculated with CA-125 (0.680). Ovarian mCRC and primary OC can be distinguished based on patient anamnesis, ultrasound pattern recognition, a proposed decision tree model, and an ADNEX model with CA-125 levels.

## 1. Introduction

Colorectal cancer (CRC) is the second most common cancer in women worldwide [1]. Approximately 50–60% of patients with CRC develop metastatic disease (mCRC), most frequently metachronously [2]. Typically, mCRC involves the liver, lung, peritoneum, lymph nodes, or ovary, and is confined to a single or few organs [3]. Approximately 17% of patients with mCRC have peritoneal carcinomatosis, while the peritoneum is the only site of metastasis in just 2%. Patients with peritoneal carcinomatosis have a shorter progression-free survival and overall survival (OS) than those without peritoneal involvement [2]. 

The National Comprehensive Cancer Network colon cancer guidelines state that patients with potentially resectable mCRC should undergo upfront evaluation by a multidisciplinary team and surgical resection, if feasible [2]. Long-term survival can be attained in 20%–50% of patients who undergo complete resection [3]. The clinical prevalence of ovarian metastases among patients with CRC was 6.4%, and 5-year OS for those with ovarian and extra-ovarian metastases was 24.7% and 34.5%, respectively (difference not significant) [4]. OS was longer if surgical resection of ovarian metastases was performed, compared with chemotherapy alone (43.1 vs. 17.0 months) [5].

In a cohort of female patients with peritoneal mCRC who underwent cytoreductive surgery and hyperthermic intraperitoneal chemotherapy (CRS-HIPEC), those with ovarian metastasis had similar outcomes to those without ovarian involvement [6]. Thus, for patients with peritoneal and ovarian CRC metastases, complete CRS-HIPEC may facilitate prolonged survival and should be attempted in experienced high-volume centers [3,6]. 

According to ultrasound imaging studies, metastatic tumors originating from CRC are the most frequent secondary malignancies, located in the pelvis (34.8%) [7] or ovaries (21–29.4%) [8,9,10,11]. Ovarian metastases derived from the colon, rectum, appendix, or biliary tract differ from those derived from the stomach, breast, lymphoma, or uterus [7,8,9]. Various characteristics of CRC metastases to the pelvis differ from those of primary ovarian cancer (OC) [7]. 

Subjective ultrasound assessment (SA) or the assessment of different neoplasias in the adnexa (ADNEX) model can be used to raise suspicion of metastatic lesions in women with suspected pelvic tumors [10].

It is important to differentiate primary and metastatic cancers during imaging, in order to facilitate optimal pretreatment work-up, multidisciplinary consultations, and treatment. 

This study aimed to identify the clinical and ultrasound characteristics that discriminate ovarian mCRC from primary OC. 

## 2. Materials and Methods

### 2.1. Study Design and Participants

This was a multi-institutional, retrospective, observational study, including patients who underwent expert ultrasound examination, with stored images and videoclips (made from 1996–2018), and in whom ovarian metastasis from colorectal and appendiceal cancer was confirmed by histological examination of a surgically excised tumor or Tru-cut biopsy. Cases with insufficient ultrasound data, those in whom suspicion of metastatic cancer was based on computed tomography or magnetic resonance imaging findings, those with metastases from primary tumors other than CRC, and those with metastases to non-ovarian pelvic organs were excluded from this study. 

The control group included patients with primary epithelial OC (confirmed by histological examination of surgical specimens; borderline ovarian tumors or non-epithelial tumors were not allowed) who underwent expert ultrasound examination (using the same scan protocol as the study group) and who were matched to patients with ovarian mCRC by age. For every patient with metastatic cancer, one patient with primary OC was provided by each participating center. A control cases with epithelial OC included next consecutive patients (matched by age) examined after ones with ovarian mCRC. 

### 2.2. Ultrasound Examination

Each patient underwent ultrasound examination, performed by a sonologist experienced in gynecologic oncology, using several commercially available machines, with both transvaginal and convex (transabdominal) probes with B and Doppler modes. Each researcher reviewed their images and videoclips using terminology provided by the International Ovarian Tumor Analysis Group (IOTA) [12], with additional sonographic features: tumor mobility, ovarian crescent sign, signs of necrosis, and whether the solid papillary projection in a cystic tumor originated from the inner wall or septa, as described elsewhere [7,8,9] and listed in Table 1. Detailed reports from the examination were also allowed if they followed the IOTA standard examination protocol for ovarian mass evaluation and description terminology, and if they included additional desired data. There was no central verification of images or videos. 

Ultrasound signs of metastases were sought using transvaginal and transabdominal probes. If signs were present, they were classified (as described elsewhere [13]) as parenchymal in the liver or spleen; carcinomatosis (inner abdominal or pelvis wall, on bowel wall or mesentery, on liver surface, etc.); omental involvement; bowel mesentery retraction; and other (including frozen pelvis, enlarged retroperitoneal lymph nodes, abdominal wall tumor, and spleen hilum involvement).

The mobility of tumors was determined by observing videoclips, or was taken from the original ultrasound report when the data were reported. The mobility of tumors was defined by their movement with respect to adjacent structures when pressed with the probe or the examiner’s hand, with simultaneous transvaginal or transabdominal ultrasound scanning. A tumor was considered mobile when it moved freely all around its perimeter in relation to the adjacent structures; semi-fixed if it was firmly attached by at least part of its perimeter, or if the adjacent structures did not show any sliding; and fixed if it was completely immobile [7] (Video S1). An ovarian crescent sign was considered present if a normal ovary sonographic structure was detected at the borders of the tumor (Appendix A). Ultrasound signs of necrosis were defined as heterogeneous, avascular areas of mixed echogenicity, with blurred borders radiating to adjacent vascularized tissue [7]. 

We collected the following clinical data: patient age; menopausal status (postmenopausal was defined as more than one year of amenorrhea without diagnosis of any endocrine disease that could influence menstrual cycles, receiving hormonal replacement therapy for menopausal symptoms, or ≥50 years of age with a previous hysterectomy); history of CRC treatment (yes/no and, if yes, whether it was surgery, chemotherapy, radiotherapy, or a combination of therapies, and time since that treatment); and (optionally) levels of serum cancer markers, including cancer antigen 125 (CA-125) and carcinoembryonic antigen (CEA), before surgery/ovarian lesion biopsy. 

Of the many available multivariable ultrasound-based models, only ADNEX can be used to raise suspicion of metastatic lesions; thus, this model was tested with respect to its performance for the discrimination of ovarian mCRC from primary OC. ADNEX calculations were conducted using the IOTA web application (Available online: https://www.iotagroup.org/sites/default/files/adnexmodel/IOTA%20-%20ADNEX%20model.html), and both percentage (%) and relative risk (RR) values were determined for the risk of malignancy and risk of metastatic tumor (ADNEX-meta). 

Diagnoses suggested by ultrasound examiners before ovarian surgery/biopsy were classified based on SA by the examiner, and were divided into five categories (Table 2).

Ultrasound and clinical data were entered into a shared Google Sheet database by each researcher. No patient-identifying data were collected. Each case received a unique identifier consisting of the institution’s name and a consecutive number. For the control group, a letter P (indicating primary) was added.

A time interval between ultrasound examination and surgery/biopsy of up to 90 days was permitted for patient inclusion in the study.

The reference standard was a histological examination with immunohistochemical evaluation, in cases with inconclusive hematoxylin and eosin testing of surgically removed ovarian tumors or tissue obtained by an ultrasound-guided Tru-cut biopsy. 

The reference standard (histology) results were not available to the ultrasound examiners performing the original examination. However, at the re-review of stored ultrasound reports, images, and videoclips, undertaken as part of this study, readers were aware of the histology.

### 2.3. Analysis 

A Student’s *t*-test and Fisher’s exact test were used to assess the significance of differences between continuous and categorical variables, respectively. The ADNEX model was created to predict five possible outcomes [14]. However, in the current study, there were only two groups: ovarian mCRC and primary OC. Therefore, the area under the receiver–operating characteristics curve (AUC) was calculated based on the conditional risk of ovarian mCRC, which was obtained by calculating risk(metastatic)/(risk(metastatic) + risk(primary OC)) [15]. We calculated the difference in AUC for ADNEX for risk of metastatic cancer as a percentage values (ADNEX-meta-%), and ADNEX for risk of metastatic cancer presented as a relative risk value (absolute predicted risk divided by baseline risk [16]; ADNEX-meta-RR). We also applied three approaches to test ADNEX for its power to predict mCRC versus primary OC. First, median values (% and RR) for risk of metastases were compared. Second, the proportions of patients with higher absolute values of ADNEX-meta than ADNEX-primary were determined. Third, the proportions of patients with predefined RR thresholds >3 and >4 were determined (as suggested elsewhere [16]). 

The performance of SA was determined with comparisons of both groups, according to predefined possible indications (Table 2); choosing one of these indications was obligatory for every case. For the AUC calculations, the SA indication of a metastatic tumor was opposed to other SA indications. Differences in sensitivity were calculated when specificity was fixed to the level obtained for subjective examiner diagnosis (and vice versa).

A classification and regression tree (CART; decision trees) procedure for the differentiation of ovarian mCRC from primary OC was proposed. CART is a graphic method for supporting the decision-making process, which can be used for different types of modeled variables, i.e., continuous or discrete. The goal is to create a model that predicts the value of a target variable based on several input variables [17,18]. The main task of the CART method is to generate mutually exclusive regions in which as many samples as possible are classified into one group. These regions are created by successive divisions of the training set using binary logic rules [19]. The learning process is carried out to obtain the most homogeneous group of sample sets (Appendix A). As an algorithm output, CART can provide two types of information: the description of which group the examined object is located in, or the probability of belonging to a given group [20]. The main advantage of using decision trees is that they are nonparametric methods. An additional advantage is the automatic identification of the most significant variables by the algorithm and the elimination of statistically insignificant variables. Among the disadvantages, notably, a slight modification of the training set (e.g., removal of several observations, change of variables thresholds) can radically change the structure of the tree. In addition, in each step, the tree can divide the space only in relation to one variable (in other words, the dividing lines are always perpendicular to the divided axis in the variable space) [21].

All clinical and sonographic data, except for cancer markers, were used as input data to build the CART. The aim was to create a clinically significant (likely to impact medical practice [22]) decision tree with a power of 75% or more (25% probability of making a type II error: false negatives). Based on subjective matter knowledge that metastatic ovarian tumors derived from colorectal cancers are similar, to some degree, to primary ovarian cancers, and with a retrospective design of the study, we aimed at a power of 75% or more. To obtain such a decision tree, we proceeded as follows. First, we used different threshold values for each variable at input, based on subjective matter knowledge [23]. Second, we used different fractions of training and test sets (70%/30%, 75%/25%, 80%/20%, 85%/15%) using a random split. Both of these procedures enabled us to choose the optimal tree structure. Simple validation was performed on the test set to evaluate classification correctness (the AUC being the validation parameter). The CART is not a mathematical model (there is no equation), so no other methods of validation (e.g., cross-validation) were used, because every modification of the tree would change the tree structure completely—there would be different results for different tree structures. Thus, there was no rationale for primary model mistake estimation, other than simple validation.

Statistical analyses were performed using RStudio 1.1.463 software (RStudio, Inc., Boston, MA, United States) [24] in the R language (R Foundation for Statistical Computing, Vienna, Austria) [25]. The ROSE (Random Over-Sampling Examples) [26] and rpart [27] libraries were used for calculations. There were no indeterminate index test or reference standard results.

Where important ultrasound data (e.g., variables from the IOTA terminology) or reference test results were missing, cases were excluded from the analysis. If cancer antigen results were missing, cases were included, but if calculations needed a marker (e.g., ADNEX), then the analysis was conducted using only cases with available data. ADNEX model calculations were performed only for cases with CA-125 marker data available, and results were compared between the same number (1:1) of patients with metastatic and primary cancers, matched by age. Additionally, we performed the same analysis on the same groups, but with ADNEX calculated without CA-125 (all clinical and ultrasound data were not changed, while the marker value was not inserted into the calculation). Both percentage (%) and RR results from the ADNEX analysis were evaluated and compared between groups. Model calculations were conducted without knowledge of the final histology data, as test and control groups were recorded in a single database, with final histology data blinded for model calculations. Intended sample size was not determined before the analysis. Standards for Reporting Diagnostic Accuracy Studies (STARD) guidelines [28] were followed.

## 3. Results

In total, 81 patients with ovarian mCRC and 81 with primary OC were identified from eight institutions (seven oncology centers). Details of patient characteristics and ultrasound parameters are presented in Table 1, Table 3, and Figure 1.

Ovarian mCRC histology indicated adenocarcinomas in all cases. Forty-nine (60%) patients were diagnosed with colon cancer at the time of surgery or biopsy of the ovarian metastatic tumor. Thirty-two (40%) patients with metastatic ovarian cancer had previously undergone treatment for CRC, with a median time of 20 months before diagnosis with adnexal mCRC; the majority had also previously undergone colorectal surgery (30/32) and chemotherapy (31/32).

The most frequent histologic type of primary OC was serous adenocarcinoma. None of the patients with primary OC had been treated for CRC, nor was CRC a synchronous malignancy.

Metastatic tumor histology samples were obtained from adnexectomy specimens and Tru-cut biopsies in 63 (78%) and 18 (22%) cases, respectively. All primary OC tumors were obtained by surgery. Patients with primary OC were diagnosed as International Federation of Gynecology and Obstetrics (FIGO) stage I (*n* = 10), II (*n* = 5), III (*n* = 52), or IV (*n* = 14).

The median time from ultrasound examination to surgery or biopsy was one day (range 0–52) and one day (range 0–90 days) for patients with metastatic and primary cancers, respectively.

### 3.1. Ultrasound

Upon ultrasound examination, ovarian mCRC were mostly solid (43%), multilocular-solid (45%), unilateral (61%), or mobile or semi-fixed (70%), with the median largest tumor and solid component diameters of 94 and 67 mm, respectively. Necrosis was noted in 49% of lesions (Figure 2). A layered structure was noted in tumors with cystic components (Figure 3), most of which had more than five locules (Appendix A, Video S2) and solid papillary projections growing from the septa, or the septa and inner wall (Figure 4). Most tumors were vascularized (84%), with moderate to marked blood flow detected in 53% of cases. On abdominal ultrasound scans for disseminated disease, metastases were detected in 46% of cases and classified as omental cake (46%), peritoneal carcinomatosis (46%), parenchymal liver metastases (27%) (Appendix A), and other (32%). Ascites was detected in 32% of women.

After the collection of cases for the study was closed, a new observation was noted (by M.S.) when analyzing the videoclips of ovarian mCRC in detail. A tree-like sign was detected in multilocular-solid tumors, with parallel, closely-localized, septa (“trunk”) later branching in different directions (“branches”), forming an image resembling a tree (Figure 5, Video S3). Interestingly, the tree-like sign was not detected in any of matched controls with primary OC.

Ovarian mCRC tumors were significantly larger, a necrosis sign was more often present, and tumors had an irregular wall or were fixed less frequently; ascites, omental cake, and carcinomatosis were less common in mCRC than primary OC (Table 1). In a subgroup of patients with ovarian mCRC who had not undergone treatment for CRC in anamnesis, tumors were larger and had fewer papillations and more locules compared with primary OC; these patients also had abdominal metastases detected with an ultrasound less often than patients with primary OC (Figure 6).

### 3.2. Serum Markers

CA-125 data were available for 66 and 79 patients with ovarian mCRC and primary OC, respectively. The median level of CA-125 was significantly higher in patients with primary OC than in those with mCRC (Table 3).

CEA data were available for 46 and 17 patients with metastatic and primary cancers, respectively. Median CEA values were significantly higher and CA-125/CEA ratios were lower among patients with metastasis, relative to those with primary cancers (Table 3). Among patients with ovarian mCRC, 18/46 (39.1%) had CEA levels <5 ng/mL, and in 14/46 (30.4%) patients they were <2.5 ng/mL. The AUC for correct classification of metastatic tumors based on a CA-125/CEA ratio cut-off value of <25 was 0.69, whereas cut-off values of <100 or <120 yielded AUC vales of 0.84 and 0.88, respectively.

### 3.3. Predicting Ovarian mCRC

Detailed results of the performance of the ADNEX model and SA are presented in Table 2, Figure 7 and Figure 8, and Appendix A. The highest AUC was for the decision tree model (0.768; see below), followed by SA (0.735) and ADNEX risk of metastasis calculated with CA-125 (0.680 (for %) and 0.678 (for RR)). The ADNEX risk of metastasis calculated without CA-125 failed (AUC = 0.512) to discriminate secondary from primary cancers.

We evaluated the performance of ADNEX using three approaches. First, for calculations with CA-125 (*n* = 66), we found significant differences between ADNEX-meta-% and ADNEX-meta-RR for patients with mCRC and primary OC. For calculations without CA-125 (*n* = 66), there were no differences between patients with mCRC and primary OC in the results generated using these models (Table 2). Second, in the group with ovarian mCRC and CA-125 data available (*n* = 66), ADNEX-meta-% was higher (indicating metastatic origin) than ADNEX-primary-% (stage I or II–IV) in only three cases (4.5%) (pattern recognition: these tumors were solid, 33–80 mm in diameter, with a color score of 3, and signs of necrosis, ascites, and metastases in the abdominal cavity in two cases; CA-125 levels ranged from 8 to 18 U/mL). Again, in the group with ovarian mCRC and CA-125 available, ADNEX-meta-RR was higher (indicating metastatic origin) than ADNEX-primary-RR (stage I or II–IV) in 19 cases (28.8%) (pattern recognition: these tumors were solid (74%) or multilocular-solid (26%), 33–160 mm in diameter, with color score of 3–4 (74%), and signs of necrosis (74%), ascites (42%), and metastases in the abdominal cavity (53%); CA-125 levels ranged from 6 to 185 U/mL; among these patients, SA was always malignant, although suspicion of metastatic and primary tumors was often equal). In the same group of 66 patients with ovarian mCRC, we calculated the ADNEX model without CA-125—no patient had an ADNEX-meta-% value higher than the ADNEX-primary-% (stage I or II–IV; none would be suspected for metastatic origin), while eight patients (12.1%) had higher ADNEX-meta-RR than ADNEX-primary-RR (stage I or II–IV) values (only this fraction of patients would be suspected of having metastatic tumors). Third, the proportions of patients with ovarian mCRC and ADNEX-meta-RR results exceeding predefined thresholds (>3 or >4) were determined, and are shown in Table 4.

### 3.4. Decision Tree

The optimal decision tree was developed and validated using data from 162 patients, randomly divided into training (80%) and test (20%) sets. At input, all clinical and ultrasound data were used except cancer markers. The resulting tree is shown in Figure 6. For example, a patient with an adnexal mass who underwent CRC treatment in the past should be directly suspected of having a metastatic tumor (irrespective of ovarian tumor sonographic features). A patient without CRC in anamnesis who has a tumor of <40 mm should be suspected of having primary OC (0% risk of mCRC). A patient with an ovarian tumor of ≥40 mm with more than three papillations has an 88.9% probability of primary OC and an 11.1% risk of mCRC. A patient with a tumor of ≥40 mm, <3 papillations, and ≥6 locules has a 91.7% probability of having an ovarian mCRC, while those with <6 locules and metastases detected in the abdominal cavity should rather be suspected of having primary OC, although the risk of metastatic cancer remains at 29.6%. The AUC of this decision tree was 0.768.

## 4. Discussion

In this study, previous treatment for CRC was the strongest clinical variable; if this is present, and there is an ovarian tumor, metastasis should be suspected, regardless of ultrasound parameters. All but one patient who had previously had CRC underwent chemotherapy, and hence had high-risk disease at diagnosis. On the ultrasounds, ovarian mCRC was mostly solid or multilocular-solid. Compared with primary OC, mCRC tumors more often had signs of necrosis in the solid components; less often, they had irregular internal walls, if locular, and were more mobile. Those with ovarian mCRC who had not previously undergone treatment for CRC had larger tumors with fewer papillations and more locules compared with primary OC tumors. Upon abdominal scanning for cancer dissemination, patients with OC had ascites, peritoneal carcinomatosis, and omental involvement more often, and liver metastases less often than those with mCRC. In addition to the previously described layered structure of a multilocular metastatic tumor [7], an ultrasound image resembling a tree was noted. Primary OC and mCRC are graphically compared in Figure 9. The most accurate methods for discriminating ovarian mCRC from primary OC were the proposed decision tree model and SA, which both generated fair AUC values. Application of the ADNEX model in this patient population resulted in a poor AUC value when CA-125 was included, and failed without CA-125. Moreover, the absolute percentage and RR values for risk of metastases were higher than those at risk of primary OC; however, it is not clear whether these differences are clinically significant.

According to other authors, among patients diagnosed with metastatic ovarian tumors, the prevalence of previous treatments for non-gynecologic malignancy was between 25–27% [7,10] and 43% [8]. In a small set of patients with history of CRC (*n* = 6), recurrence in ovaries were diagnosed in four cases, and metachronous primary OC in two cases [10].

In the largest published cohort of patients with ovarian metastatic cancers who underwent comprehensive ultrasound examination, the median tumor diameter was 86 mm; the majority (95%) had a solid component, with a median proportion of solid tissue of 100% (range 64–100) [14]. Based on ultrasound and clinical variables, the main factors associated with metastatic pelvic tumors were younger age, lower serum CA-125, previous neoplasia, and mobile tumors, whereas the presence of omental involvement was associated with primary OC [10]. Carcinomatosis was detected in 25% of patients, all in combination with ascites (present in 86%); liver metastases were observed in 30% of patients. Ultrasound signs of necrosis were observed in more than half of the metastatic pelvic tumors [7].

According to published data, the appearance of ovarian/pelvic CRC metastases on ultrasounds differed from that of tumors of other origins: tumors were less often purely solid, median tumor diameter was significantly larger, irregular external borders and papillary projections were more common, and tumors appeared less vascularized [7,8,9]. Purely solid tumors accounted for between 18% and 47% of ovarian mCRCs [7,8,9], bilateral lesions were found in 6%–18% of cases, median CA-125 was 89.3 U/mL, and patients were diagnosed at various ages (31–96 years) [7,9].

According to Zikan et al. [7] pelvic cystic metastases from CRC (*n* = 32) had a layered structure, with papillary projections and necrosis in the hypoechogenic solid portion (90.6%). Papillary projections growing from thin septa were observed in 15.6% of tumors. Carcinomatosis was detected (in combination with ascites) in only 6.3%, and liver metastases in 18.8%, of cases [7].

The ADNEX model has previously been reported to show fair (AUC = 0.71) and good (AUC = 0.82) discrimination between stage I or II–IV primary OC and secondary malignant tumors, respectively, with the latter being largely attributable to the inclusion of CA-125 levels in the model. Application of the ADNEX model without inclusion of CA-125 mainly affected discrimination between stage II–IV primary and metastatic OC, where AUC decreased from 0.82 to 0.59 [14]. Owing to the low prevalence of most malignant outcome categories, the predicted risk for metastases to the ovaries was not always very high. The baseline risk for this group was only 4%, whereas the maximum predicted risk was 44% [16]. Practical guidance for applying the ADNEX model suggests that the model indicates an increased risk of metastatic cancer if RR is >3 or >4 [16]. In a study that aimed to differentiate disseminated metastatic from primary OC, SA predicted primary OC with a sensitivity of 82% and a specificity of 70%; the ADNEX model and CA-125 had comparable accuracy [10].

Ovarian tumors may be suspected to be of gastrointestinal origin, particularly when the CA-125 U/mL/CEA ng/mL ratio is ≤25 [29]. More et al. studied it in greater detail—in a subset of patients with multilocular and multilocular-solid tumors (based on ultrasound), the best cut-off values for CEA alone and the CA-125/CEA ratio for discrimination between ovarian primary (including benign and malignant) and metastatic tumors were 2.3 and 11.9, respectively [11].

The strengths of this study are inclusion of the largest number (to our knowledge) of patients with ovarian mCRC analyzed with a standardized ultrasound protocol; collections of data from multiple institutions; inclusion of clinical and laboratory data; and comparisons with patients with primary OC. Moreover, both raw parameters and the performance of available predictive models were compared. Our new approach, using a decision tree model together with example figures, videoclips, and a diagram, represents a comprehensive study of ovarian mCRC.

The study has also some limitations. First, it was retrospective, although the protocol only allowed inclusion of cases with documented, standardized examination. Second, the ultrasound examination was performed by sonologists experienced in gynecologic oncology; thus, some results may not be applicable to less experienced operators. Nevertheless, if well-described terms and definitions for ultrasound are adhered to, our results could be generalized. Also, results of SA should be interpreted with caution, because the predefined indications might introduce a bias, e.g., “malignant, not specified” might include secondary as well as primary cancers, and “malignant, metastatic” did not specify the origin of primary cancer. Third, CEA levels were unavailable for many patients; hence, results based on this variable and the CA-125/CEA ratio should be interpreted with caution. Fourth, the proposed decision tree model may not be applicable to patients who have previously undergone treatment for low-risk CRC, and are now presenting with an ovarian tumor. Also, no family medical history or genetic risk information was available.

## 5. Conclusions

Metastasis of CRC to the ovaries may be predicted based on patient anamnesis and ultrasound pattern recognition. Our proposed decision tree model or an expert ultrasound assessment would be a more objective method to differentiate CRC metastases from primary OC. The ADNEX model could be used if the CA-125 level is included; however, the results should be interpreted with caution. Comprehensive clinical and ultrasound diagnostic work-up should assist clinicians in decisions about optimal pretreatment management, which could involve ultrasound-guided, Tru-cut biopsy and referral for colonoscopy, followed by multidisciplinary evaluation if mCRC is suspected. A prospective trial is needed to test our results and the model.

## Figures and Tables

**Figure 1 diagnostics-09-00210-f001:**
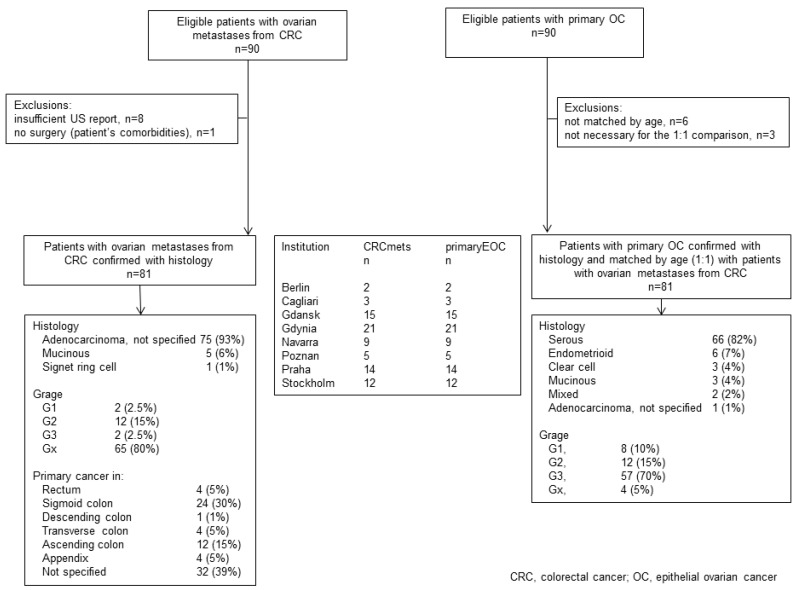
Flow of participants.

**Figure 2 diagnostics-09-00210-f002:**
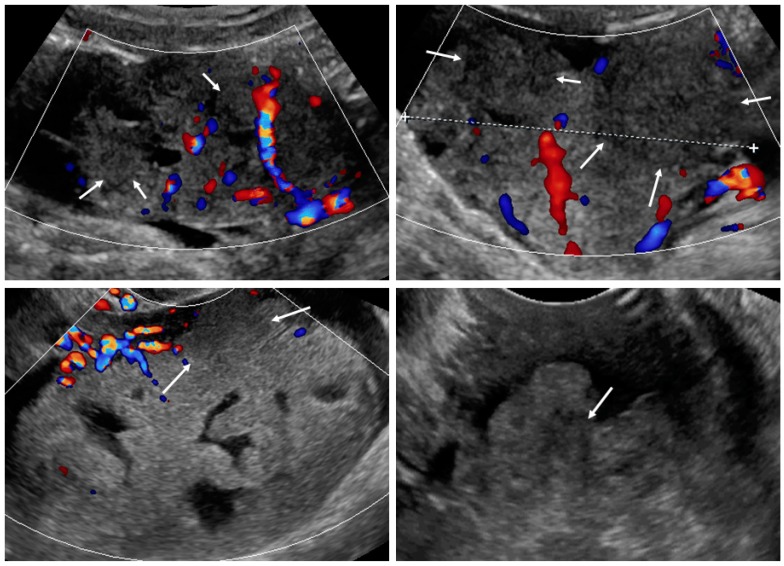
Necrosis sign [7] in tumor solid portions. Arrows indicate area suspected for necrosis. White dotted line is a measurement of the tumor.

**Figure 3 diagnostics-09-00210-f003:**
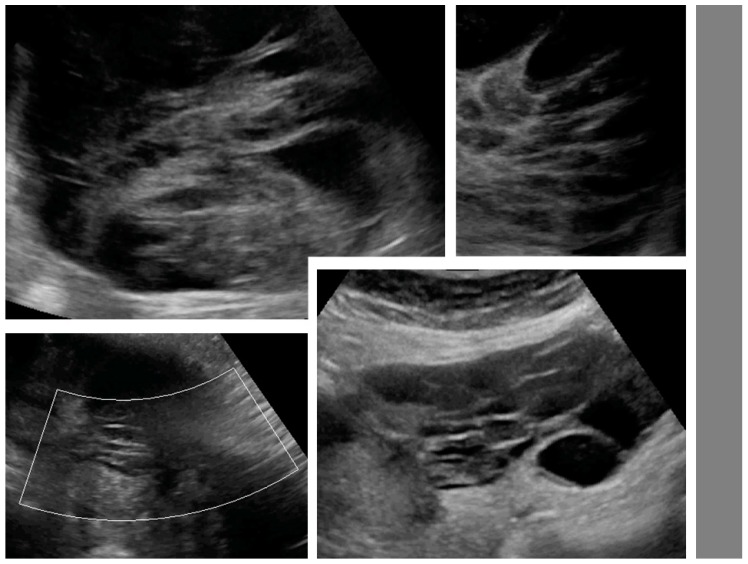
Layered structure of multilocular tumors.

**Figure 4 diagnostics-09-00210-f004:**
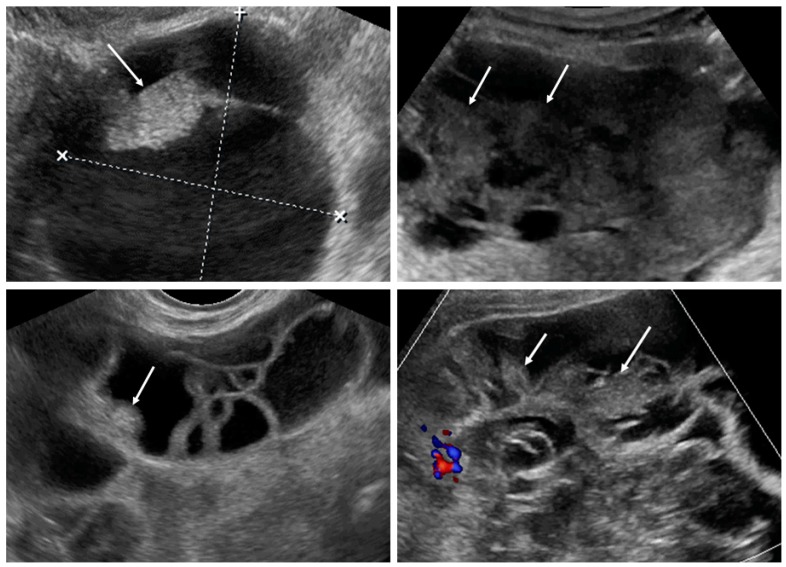
Solid papillary projections (arrows) growing from the septa and inner wall. White dotted lines are measurements of the tumor.

**Figure 5 diagnostics-09-00210-f005:**
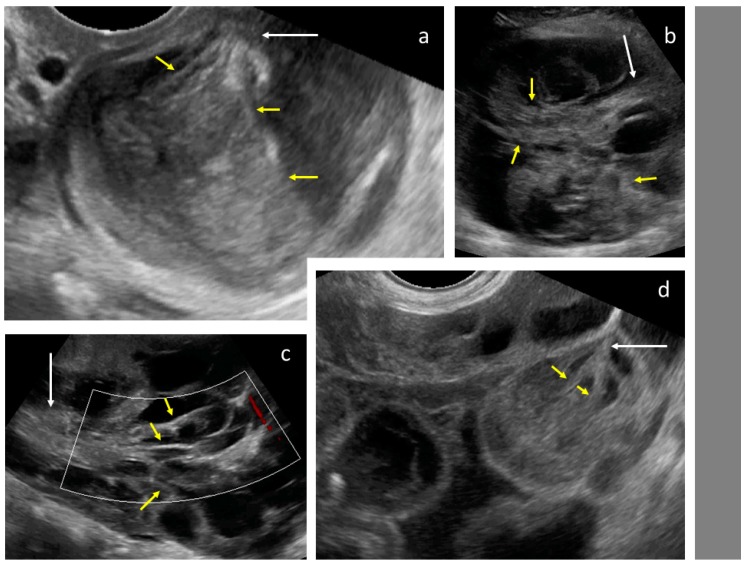
Tree-like sign in a multilocular-solid tumor. White arrows indicate the “tree trunk”; yellow arrows indicate “tree branches”.

**Figure 6 diagnostics-09-00210-f006:**
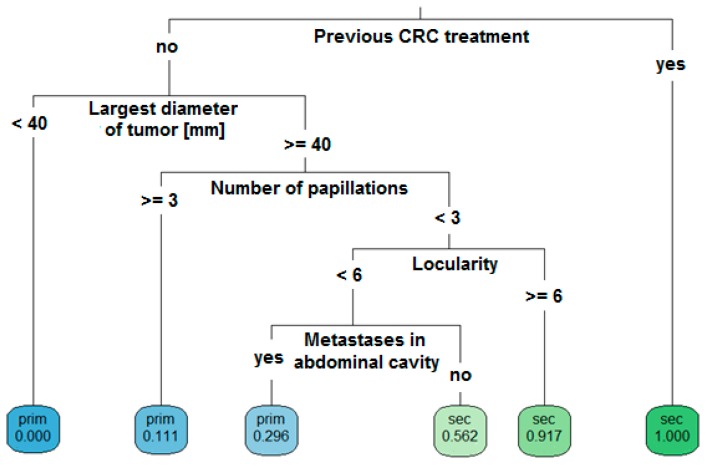
Decision tree model. Ultrasound-based, step-by-step, differential diagnosis of ovarian metastatic colorectal cancer (sec) and primary cancer (prim) for patients with a complex adnexal mass.

**Figure 7 diagnostics-09-00210-f007:**
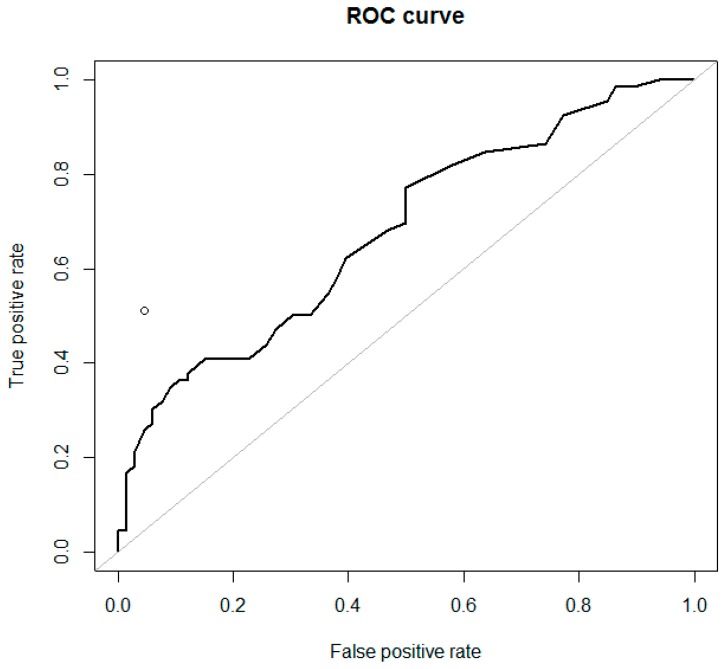
Receiver-operating characteristic (ROC) curves for the ADNEX model (RR), calculated with CA-125 (*n* = 66) to differentiate ovarian metastatic colorectal cancer from primary cancer in women with an ovarian tumor (AUC = 0.678). The performance of subjective ultrasound assessment is plotted in the ROC space (○) (AUC = 0.735).

**Figure 8 diagnostics-09-00210-f008:**
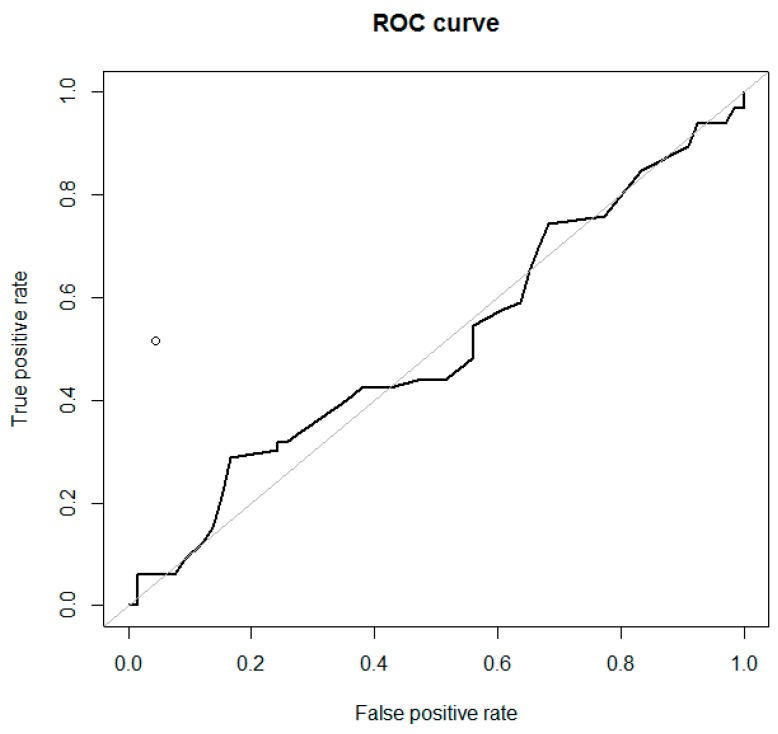
Receiver-operating characteristic (ROC) curves for the ADNEX model (RR), calculated without CA-125 (*n* = 66) to differentiate ovarian metastatic colorectal cancer from primary cancer in women with an ovarian tumor (AUC = 0.511). The performance of subjective ultrasound assessment is plotted in the ROC space (○) (AUC = 0.735).

**Figure 9 diagnostics-09-00210-f009:**
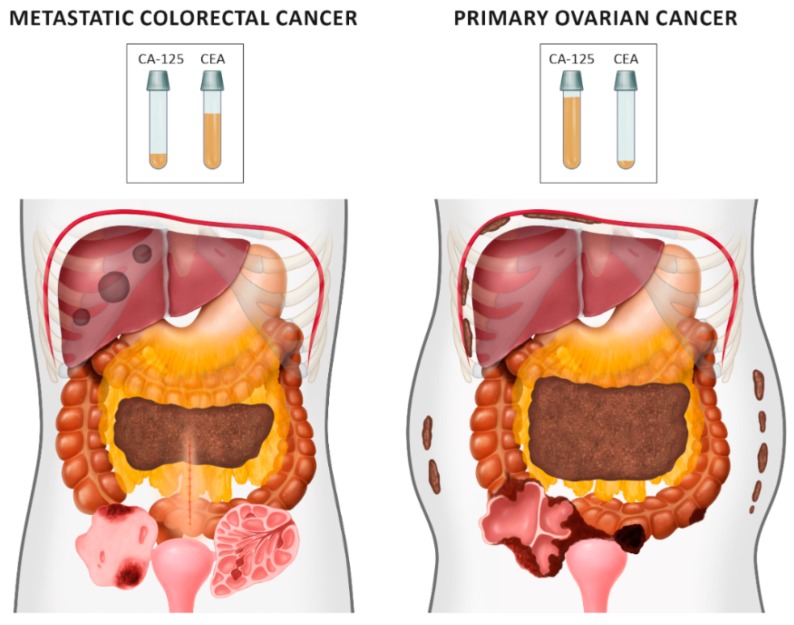
Differences between ovarian metastatic colorectal cancers (left) and primary cancers (right), based on ultrasound and clinical examinations.

**Table 1 diagnostics-09-00210-t001:** Ultrasound parameters of ovarian mCRC and primary OC, obtained by transvaginal and abdominal scanning.

Characteristics	Ovarian mCRC	Primary OC	*p*
	*n* (%)	*n* (%)	
Laterality			0.241
Unilateral	49 (60.5%)	44 (54.3%)	
Bilateral	32 (39.5%)	34 (42%)	
Not possible to determine	0	3 (3.7%)	
Largest diameter of the tumor (mm), median (range)	94 (33–350)	90 (24–300)	0.129
Largest diameter of the solid component (mm), median (range)	67(0–160)	52 (0–220)	0.052
Type of tumor (structure)			0.913
Unilocular	0	0	
Multilocular	2 (2.5%)	1 (1.2%)	
Unilocular-solid	6 (7.4%)	9 (11.1%)	
Multilocular-solid	38 (46.9%)	34 (42%)	
Solid	35 (43.2%)	37 (45.6%)	
Echogenicity (of a cystic component)	*n* = 46	*n* = 45	0.754
Anechoic	5	9	
Low level	16	7	
Ground glass	1	1	
Hemorrhagic	0	5	
Mixed	24	23	
Locularity			0.253
0	34 (42%)	38 (46.9%)	
1–5	21 (25.9%)	27 (33.3%)	
6–10	16 (19.8%)	7 (8.7%)	
>10	10 (12.3%)	9 (11.1%)	
Number of solid papillary projections			0.07
0	59 (72.9%)	49 (60.5%)	
1	3 (3.7%)	8 (9.9%)	
2	4 (4.9%)	2 (2.5%)	
3	6 (7.4%)	7 (8.6%)	
>3	9 (11.1%)	15 (18.5%)	
Solid papillary projections originating from			0.154
Septa	7 (8.6%)	3 (3.7%)	
Inner wall	5 (6.2%)	15 (18.5%)	
Septa and inner wall	8 (9.9%)	10 (12.3%)	
Not possible to determine	2 (2.5%)	4 (4.9%)	
No papillary projections	59 (72.8%)	49 (60.6%)	
Blood flow in papillary/solid component (yes)	68 (84%)	63 (77.8%)	0.321
Irregular wall (yes)	46 (56.8%)	61 (75.3%)	0.013
Acoustic shadows (yes)	2 (2.5%)	8 (9.9%)	0.051
Doppler Color Score			0.552
No blood flow (1)	13 (16%)	15 (18.5%)	
Minimal flow (2)	25 (30.9%)	16 (19.8%)	
Moderate flow (3)	23 (28.4%)	27 (33.3%)	
Marked blood flow (4)	20 (24.7%)	23 (28.4%)	
Mobility (“sliding sign”) (available/all)	74/81	76/81	<0.001
Mobile	27 (36.5%)	14 (18.4%)	
Semi-fixed	25 (33.8%)	14 (18.4%)	
Fixed	22 (29.7%)	48 (63.2%)	
Ovarian crescent sign (available/all)	75/81	81/81	0.321
Yes	1	0	
Necrosis suspected (available/all)	68/81	76/81	<0.001
Yes	33 (48.5%)	13 (17.1%)	
Ascites (yes)	26 (32.1%)	41 (50.6%)	0.017
Metastases in abdominal cavity in ultrasound (yes)	37 (45.7%)	54 (66.7%)	0.007
Location (if present)			
Parenchymal, liver	10 (27%)	7 (13%)	0.117
Parenchymal, spleen	0	0	
Carcinomatosis	17 (46%)	43 (79.6%)	<0.001
Omental cake	17 (46%)	36 (66.7%)	0.039
Bowel mesentery retraction	3 (8%)	3 (5.6%)	0.668
Other	9 (24.3%)	12 (22.2%)	0.872

mCRC: metastatic colorectal cancer; OC: ovarian cancer.

**Table 2 diagnostics-09-00210-t002:** Diagnosis suggested by ultrasound examiner, and the performance of predictive models for the discrimination between colorectal metastatic and primary ovarian cancer.

Subjective Diagnosis and Multivariable Predictive Models	Ovarian mCRC	Primary OC	*p*
	*n* (%)	*n* (%)	
Diagnosis suggested by examiner (SA)			<0.001
Malignant, not specified	25 (30.9%)	19 (23.5%)	
Malignant, primary ovarian	13 (16%)	55 (67.9%)	
Malignant, metastatic	41 (50.6%)	3 (3.7%)	
Benign	0	0	
Inconclusive	2 (2.5%)	4 (4.9%)	
ADNEX calculated with CA-125 (available/all)	66/81	66/81	
ADNEX-risk of metastasis (%), median (range)	7.2 (1.3–38.4)	4.3 (0.4–32.9)	<0.001
ADNEX-risk of metastasis (RR), median (range)	1.8 (0.3–9.6)	1.1 (0.1–8.2)	<0.001
ADNEX calculated without CA-125 (available/all)	66/81	66/81	
ADNEX-risk of metastasis (%), median (range)	8.1 (1.2–20.6)	6.5 (0.5–17.1)	0.949
ADNEX-risk of metastasis (RR), median (range)	2.1 (0.3–5.2)	1.6 (0.1–4.3)	0.978

ADNEX: assessment of different neoplasias in the adnexa (IOTA model); CA-125: cancer antigen 125; IOTA: International Ovarian Tumor Analysis Group; mCRC: metastatic colorectal cancer; OC: ovarian cancer; RR: relative risk (a results from ADNEX model calculation); SA: subjective ultrasound assessment.

**Table 3 diagnostics-09-00210-t003:** Clinical characteristics of patients with ovarian mCRC and primary OC.

Characteristics	Ovarian mCRC	Primary OC	*p*
Age, median (range) *	59.2 (30–86)	59.8 (34–86)	0.769
Postmenopausal, *n* (%)	59 (73%)	61 (75%)	0.722
Previous CRC treatment, *n* (%)	32 (40%)	0	<0.001
Time from CRC treatment to ovarian metastasis †	20 (3–60)	-	
median months (range)			
Previous CRC surgery/all treated †	30/32	-	
Previous CRC radiotherapy/all treated †	1/32	-	
Previous CRC chemotherapy/all treated †	31/32	-	
CA-125 [U/mL]			
Available/all	66/81	79/81	
Median (range)	106 (6–1300)	518 (19–21,256)	0.002
CEA [ng/mL]			
Available/all	46/81	17/81	
Median (range)	6.55 (0.3–562)	1.0 (0.2–16.5)	<0.001
CA-125/CEA			
Available/all	42/81	17/81	
Median (range)	15.6 (0.08–1200)	533 (25–13,920)	0.053

*: per study protocol, patients with primary OC were matched by age to those with metastatic disease; †: those who were previously treated for CRC; CA-125: cancer antigen 125; CEA: carcinoembryonic antigen; CRC: colorectal cancer; mCRC: metastatic colorectal cancer; OC: epithelial ovarian cancer.

**Table 4 diagnostics-09-00210-t004:** IOTA ADNEX model relative risk (RR) thresholds, with and without CA-125, in a group of patients with ovarian mCRC.

Patients with Ovarian mCRC	ADNEX with CA-125			ADNEX without CA-125		
	Relative Risk of Metastatic Cancer			Relative Risk of Metastatic Cancer		
	>2	>3	>4	>2	>3	>4
The proportion of patients with ovarian mCRC with this result (%)	43.9	31.8	21.2	50	25.8	1.5
PPV (%)	63	80.8	87.5	53.2	44.7	20

ADNEX: assessment of different neoplasias in the adnexa (IOTA model); CA-125: cancer antigen 125; IOTA: International Ovarian Tumor Analysis Group; mCRC: metastatic colorectal cancer; PPV: positive predictive value..

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
