# Peer review of "Ultrasound and Clinical Preoperative Characteristics for Discrimination Between Ovarian Metastatic Colorectal Cancer and Primary Ovarian Cancer: A Case-Control Study"

_diagnostics, 2019, doi:10.3390/diagnostics9040210_

Round 1

Reviewer 1 Report

This is a well-written manuscript that should be of considerable interest to the medical community.  I am recommending it's acceptance.  I have noted some things that are in need of correction:

Table 1 has rows out of alignment and this needs to be corrected Line 137 make correction Table 2 is out of alignment Line 179: please comment on how the predefined possible indications might introduce a bias in the study Line 200, please defend 75% power Lines 218-221, please clarify ADNEX with and without Ca125 usage Line 232, first comma Line 253, define US definition of necrosis Line 366: define what you mean by clinically useful Line 406, italics needed.  Supplementary material is excellent and a very good addition.

Author Response

Response to Reviewer 1.

Thank you very much for the review. We appreciate your work and time devoted to perform the review. Thank you for valuable comments and suggestions.

Table 1 has rows out of alignment and this needs to be corrected – corrected

Line 137 make correction – corrected

Table 2 is out of alignment – corrected

Line 179: please comment on how the predefined possible indications might introduce a bias in the study – a comment was added in lines 424-427.

Line 200, please defend 75% power – the text was added in lines 201-204.

Lines 218-221, please clarify ADNEX with and without CA125 usage

We added one information in line 222-225.

More detailed clarification is below:

“If cancer antigen results were missing, cases were included” – it means, that cases without CA125  were included in the study, because there were many other important ultrasound data that could be compared between groups.

“if calculations needed a marker (e.g., ADNEX), then the analysis was conducted using only cases with available data” – it means, that if CA-125 was needed to calculate results of a model, here ADNEX, than only cases with CA-125 were included. In other words, we did not “create / calculate” CA-125 if it was not available.

“We performed the same analysis on the same groups, but with ADNEX calculated without CA-125” – The ADNEX model is designed to be used with or without CA-125. It was shown in the literature that the lack of CA-125 may have significant impact on final results of the ADNEX calculation, and it was even more significant for discrimination of secondary from primary ovaria cancers (like in our case). Thus, we aimed to test performance of ADNEX with and without CA-125. In order to avoid the risk of bias, we included cases with CA-125 available only. Firstly, we calculated ADNEX with CA-125. Secondly, we calculated ADNEX without CA-125 on the same group of patients. In other words, calculation of ADNEX was repeated for the same case, once with, and then without CA-125. 

Line 232, first comma:

Here, it is a footnote of the table. Abbreviations are explained, with the abbreviation first, than comma, and followed with the explanation. “*” is a kind of abbreviation, thus comma is placed after the symbol, and before the explanation.

Line 253, define US definition of necrosis – it is explained in lines 125-126.

Line 366: define what you mean by clinically useful:

I have changed “clinically useful” into “clinically significant”, and I have added the definition and a reference. See lines 200 and 370.

Line 406, italics needed – corrected.

Supplementary material is excellent and a very good addition. – thank you.

Reviewer 2 Report

excellent study

Author Response

Thank you very much for the review. We appreciate your work and time devoted to perform the review. We are happy that you marked our study as excellent.

Reviewer 3 Report

This paper is very well written and worth to read.

Author Response

Thank you very much for the review. We appreciate your work and time devoted to perform the review. We are happy about your comments.